# Dirac cone, flat band and saddle point in kagome magnet YMn$_6$Sn$_6$

Man Li [1], Qi Wang[1], Guangwei Wang[2], Zhihong Yuan[2], Wenhua Song[1], Rui Lou[3,4,5], Zhengtai Liu [6], Yaobo Huang[7], Zhonghao Liu[6,8 ✉], Hechang Lei [1 ✉], Zhiping Yin [2 ✉] & Shancai Wang [1 ✉]

Kagome-lattices of 3$d$-transition metals hosting Weyl/Dirac fermions and topological flat bands exhibit non-trivial topological characters and novel quantum phases, such as the anomalous Hall effect and fractional quantum Hall effect. With consideration of spin–orbit coupling and electron correlation, several instabilities could be induced. The typical characters of the electronic structure of a kagome lattice, i.e., the saddle point, Dirac-cone, and flat band, around the Fermi energy ($E_F$) remain elusive in magnetic kagome materials. We present the experimental observation of the complete features in ferromagnetic kagome layers of YMn$_6$Sn$_6$ helically coupled along the $c$-axis, by using angle-resolved photoemission spectroscopy and band structure calculations. We demonstrate a Dirac dispersion near $E_F$, which is predicted by spin-polarized theoretical calculations, carries an intrinsic Berry curvature and contributes to the anomalous Hall effect in transport measurements. In addition, a flat band and a saddle point with a high density of states near $E_F$ are observed. These multisets of kagome features are of orbital-selective origin and could cause multi-orbital magnetism. The Dirac fermion, flat band and saddle point in the vicinity of $E_F$ open an opportunity in manipulating the topological properties in magnetic materials.

[1] Department of Physics and Beijing Key Laboratory of Opto-Electronic Functional Materials & Micro-Nano Devices, Renmin University of China, Beijing, China. [2] Department of Physics and Center for Advanced Quantum Studies, Beijing Normal University, Beijing, China. [3] School of Physical Science and Technology, Lanzhou University, Lanzhou, China. [4] State Key Laboratory of Surface Physics and Department of Physics, Fudan University, Shanghai, China. [5] Collaborative Innovation Center of Advanced Microstructures, Nanjing, China. [6] State Key Laboratory of Functional Materials for Informatics, Shanghai Institute of Microsystem and Information Technology, Chinese Academy of Sciences, Shanghai, China. [7] Shanghai Advanced Research Institute, Chinese Academy of Sciences, Shanghai, China. [8] College of Materials Science and Opto-Electronic Technology, University of Chinese Academy of Sciences, Beijing, China. ✉email: lzh17@mail.sim.ac.cn; hlei@ruc.edu.cn; yinzhiping@bnu.edu.cn; scw@ruc.edu.cn

The frustrated kagome lattice, made up of the geometry of corner-sharing triangles, has been studied intensively due to the magnetic frustration-induced quantum spin liquid state[1,2]. Meanwhile, the construction of topological band theory in recent years has greatly enriched in the electronic band structure of kagome lattice[3–5]. Theoretical studies show the kagome lattice systems as an ideal platform for understanding the topological states with novel topological excitations[6–8]. For example, a flat band (FB) can be constructed by completely destructive interference of Bloch wave functions in two-dimensional (2D) kagome lattice with nearest-neighbor (NN) interaction. Such FBs, just like a counterpart of the Landau level, can be characterized by a Chern number[3,7]. Representing a highly degenerate and quenched kinetic energy of electron state may give rise to the abundant exotic emergent effects, such as ferromagnetism, high-temperature superconductivity, Wigner crystal, and fractional quantum Hall effects[9–16]. Besides, the kagome lattice system shares similar topological physics with honeycomb, Dirac cone-type dispersions in the momentum-space $K$ point. Once a net magnetization and intrinsic spin–orbit coupling (SOC) are present, a well-separated non-trivial Chern bands, Chern gap and intrinsic quantum anomalous Hall effect could present when the Fermi energy is tuned properly[17,18].

The band structure studies of $3d$ transition-metal kagome compounds, which are usually strong correlated and exhibit magnetism, remain challenging albeit theoretical predictions[19–32]. A typical electronic structure of the kagome lattice is the existence of Dirac point (DP) at the Brillouin zone (BZ) corner, a saddle point (SP) at BZ boundary, and a FB over the whole BZ (Fig. 1b). Topological non-trivial electronic properties were observed in kagome lattices constituted of $3d$-transition metallic element, i.e., large intrinsic anomalous Hall effect originating from the Berry curvature of Dirac

cone-type dispersion of $K$ point both in ferromagnetic (FM) and antiferromagnetic (AFM) materials[18,20–24]. Another important feature is the dispersionless electronic structure in magnetic kagome materials[25,30–32] and in paramagnetic (PM) kagome materials[27–29]. In PM kagome materials, extremely flat bands close to $E_F$ have been reported in CoSn[28,29] and YCr$_6$Ge$_6$[27]. However, in magnetic systems, the direct evidence of the FBs is either unobserved (Fe$_3$Sn$_2$, Co$_3$Sn$_2$S$_2$[22,30]), or far away from $E_F$ (FeSn[25]), due to the strong correlation of $3d$ electrons, the complexity with magnetic structure, and the interplay between electron correlation and topological properties. Thus, searching for the flat band, and saddle point, in addition to the Dirac fermion near $E_F$ in magnetic kagome systems serves as the main objective to manipulate the topological properties in magnetic materials.

In this paper, we study the electronic structure of kagome lattice YMn$_6$Sn$_6$ with in-plane ferromagnetism and helical anti-ferromagnetism along $c$-axis by combining angle-resolved photoemission spectroscopy (ARPES) and density functional theory plus dynamical mean-field theory (DFT + DMFT) calculations. We report the first experimental observation of the complete characters of kagome electronic structure: Dirac cone, flat band, and saddle point, in such a magnetic system. One complete set of the characteristic of kagome lattice includes a Dirac point (DP1) above $E_F$, a saddle point (SP1) near $E_F$ and a flat band (FB1) locates at ~0.4 eV below $E_F$ across the whole BZ. They show negligible $k_z$ dispersion, suggesting the major 2D characters of the kagome structure in YMn$_6$Sn$_6$. The DFT + DMFT calculations with orbital-resolved electronic structures further confirm the 2D features with the in-plane orbital composition of $d_{xy}/d_{x^2-y^2}$. Moreover, we detected the existence of extra kagome characters, including a Dirac point (DP2) and a flat band (FB2) in the vicinity of $E_F$. The DP2 has a $d_{xy}/d_{x^2-y^2}$ with mixture of $d_{z^2}$ orbital

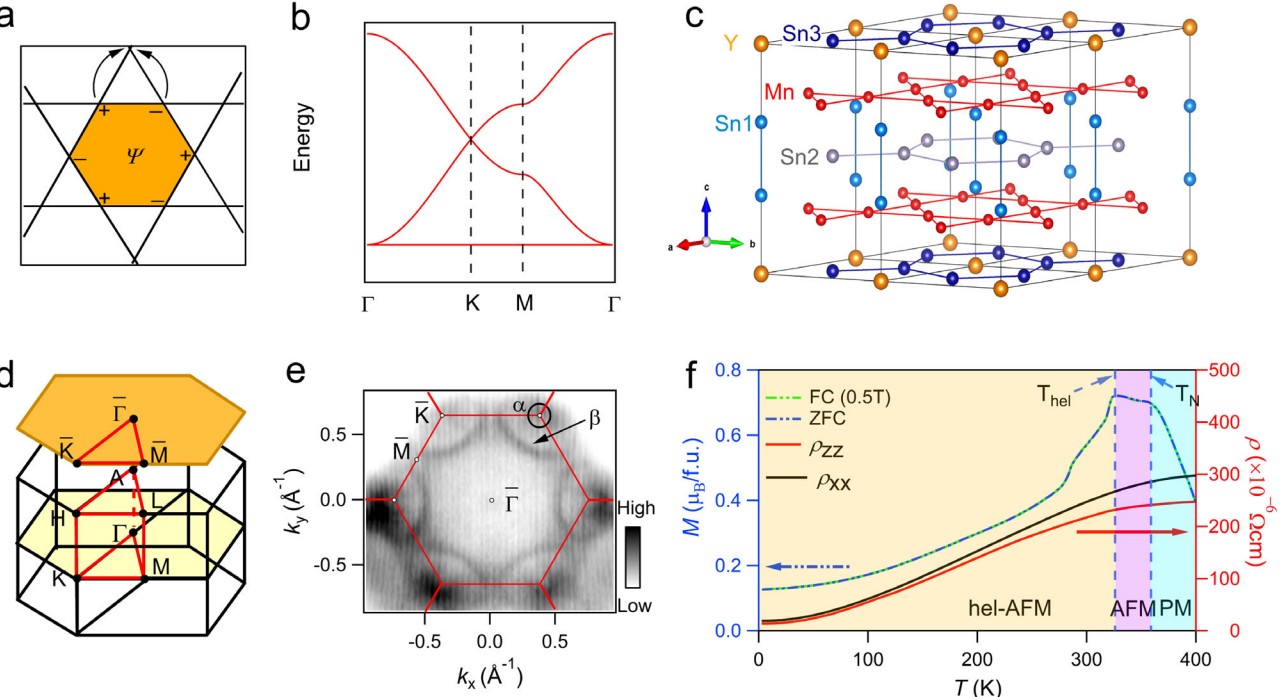

**Fig. 1 Crystal and electronic structures of YMn$_6$Sn$_6$. a** Confinement of electron eigenstate induced by destructive interference in kagome lattice with NN hopping. **b** Tight-binding calculation of band structure of kagome lattice with NN in-plane hopping without SOC, featuring the Dirac cone at the BZ corner $K$ point, a saddle point at BZ boundary $M$ point, and a FB over the whole BZ. **c** Crystal structure of YMn$_6$Sn$_6$ with space group $P6/mmm$ (No. 191). **d** 3D and projected BZs of YMn$_6$Sn$_6$ with marked high-symmetry points. **e** Photoemission intensity plot measured with 138-eV photons at $E_F \pm 10$ meV in the $k_z \sim 0$ plane. Hexagonal BZs are marked with red lines. **f** Magnetization as a function of temperature with zero field-cooling and field-cooling at $B = 0.5$ T along the [100] direction. Temperature dependence of longitudinal resistivity $\rho_{xx}$ and $\rho_{zz}$ with zero-field.

character, and presents a weak dispersion along $k_z$, while the FB2 is mainly dominated by $d_{xz}/d_{yz}$ orbitals. In the presence of SOC, the flat band and Dirac point have Chern numbers arising from the non-trivial Berry phase and supporting the orbital magnetism[32]. Altogether, these topological non-trivial bands in the kagome material will help understand the relationship between electronic/magnetic correlation and peculiar lattice geometry, and open an opportunity in manipulating the topological properties in magnetic materials.

## Results

**The crystal structure and transport property of YMn$_6$Sn$_6$.** Kagome compound YMn$_6$Sn$_6$ has a hexagonal structure with space group $P6/mmm$ (No. 191). It contains three kinds of Sn sites, stacking of Y-Sn3 layer and Mn–Sn1–Sn2–Sn1–Mn slab, with double Mn kagome lattice layers as shown in Fig. 1c. The Sn2 atoms form the honeycomb lattice located at the center of Mn–Sn1–Sn2–Sn1–Mn slab. As for the Mn kagome layers, neutron diffraction experiments reported FM coupling in each kagome bilayer, along with $c$-axis AFM coupling between the bilayer to the next bilayer below room temperature[33–35].

Figure 1d shows the 3D BZ with high-symmetry points and projected BZ along $c$-axis. In Fig. 1e, the integrated intensity at $E_F \pm 10$ meV obtained from ARPES measurement at $k_z \sim 0$ plane is shown to represent the Fermi surface (FS). There exist two pockets centered at $\bar{K}$ point as marked, both with strong matrix element effects to modulate the intensity in different BZs[36,37]. The large one ($\beta$) is clearly identified in the 1st BZ but weaker in the 2nd BZ. It looks like an "arc-like" band in the 1st BZ and crosses the zone boundary near $M$, forming a closed "triangle"-like Fermi pocket. The small one ($\alpha$) is more clearly visible in the 2nd BZ than in the 1st BZ, and forms a smaller pocket around $\bar{K}$. Figure 1f shows the magnetization as a function of temperature with zero field-cooling and field-cooling at $B = 0.5$ T along the [100] direction. Two transformative peaks at $T_N = 359$ K and $T_{hel} = 326$ K are observed. The former temperature corresponds to the paramagnetism–antiferromagnetism phase transition. The latter corresponds to a transition from an AFM order along the $c$-axis above $T_{hel}$ to a $c$-axis helical order with in-plane FM order at low temperature, consistent with the previous results[33–35,38,39]. Moreover, the temperature dependence of longitudinal resistivity $\rho_{xx}$ and $\rho_{zz}$ with zero-field shows a metallic behavior with weak anisotropy, similar to GdMn$_6$Sn$_6$[40].

**The complete characteristics of kagome structure.** In order to investigate the kagome lattice related topological electronic structure, we measured the band structures in the 1st BZ at $k_z \sim 0$ plane along with the high-symmetry lines. The intensity plots along the $\Gamma$–$M$–$K$–$\Gamma$ line and the corresponding second derivative plots are shown in Fig. 2a, b, respectively. Along $\Gamma$–$M$, one can clearly see a quadratic band (QB) with its bottom at about 0.4 eV below $E_F$ at $\Gamma$ dispersing upward towards $M$, and gradually becoming flat at the $M$ point. Along $M$–$K$, an electron-like band disperses linearly upward towards $E_F$ and acrosses $E_F$ about one third between $M$ and $K$, forming the $\beta$ FS in Fig. 1e. At $M$ point, it conforms to the dispersion of SP1 at $E_B \sim 40$ meV, as marked in Fig. 2b. From intensity plot along $\Gamma$–$K$, the $\beta$ band passes through the $E_F$, forming a large holelike FS around $K$ point, which can be more clearly seen in second derivative plots (Fig. 2b). This $\beta$ band forms a Dirac point (DP1) at about 0.3 eV above $E_F$ by linear extrapolation, as indicated by the blue circle in Fig. 2c. Another feature is a flat band at about 0.4 eV below $E_F$, FB1, which exists through the whole BZ. It can be regarded as a direct consequence of quantum phase interference effects in the kagome lattice, as shown in Fig. 1a. This flat band feature has a narrow bandwidth

and the intensity is more visualized along $\Gamma$–$K$ due to the matrix element effect. The band dispersion closely follows the DFT + DMFT calculations shown in Fig. 2c. In particular, it exhibits the complete characteristics of kagome electronic structure, a flat band (FB1) over the whole BZ touching a quadratic band at $\Gamma$ point that emerges from the Dirac band (DP1) at $K$ point and forms a saddle point (SP1) at $M$.

At the $K$ point, a linearly dispersing Dirac point (DP2) is found at around 45 meV below $E_F$, which is also characteristic of the band structure as a result of kagome lattice similar as previously observed in FeSn and CoSn[25,28,29]. According to our DFT + DMFT calculated orbital-resolved electronic structures in FM configuration, the DP2 belongs to the spin-polarized band with minority-spin state, as shown in the Supplementary Fig. 5. The DP2 can be also clearly seen from the momentum distribution curves (MDCs) and energy distribution curves (EDCs) in Fig. 2f, g, respectively. With consideration of SOC, a small bandgap <10 meV will open at DP2, adding a mass term to the linearly dispersive band, and a massive Dirac fermion thus can be formed. In consideration of in-plane FM configuration, it could realize a spin-polarized Dirac fermion with a non-trivial Chern gap in YMn$_6$Sn$_6$ as previously observed in TbMn$_6$Sn$_6$ by scanning tunneling microscopy/spectroscopy (STM/S) measurement[18]. In our result, this non-trivial Dirac fermion is in the occupied state and closer to $E_F$ than in TbMn$_6$Sn$_6$, and contributes to the intrinsic anomalous Hall effect at high magnetic field in transport measurement[34].

Figure 2 h shows a series of constant energy evolution maps in the $k_z \sim 0$. We notice that the $\beta$ band constitutes a hole pocket and the $\alpha$ band holds an electron pocket around $K$ point. Along with the energy going from $E_F$ to higher binding energy, the hole pockets gradually expand and the branches of $\beta$ band get closer to each other along $K$–$M$–$K$, finally touching at SP1 with the binding energy ~40 meV. Further away from the energy, the band forms a hole band along $\Gamma$–$M$. At $M$ point, the band dispersion conforms a saddle point at ~40 meV below $E_F$. The $\alpha$ pocket firstly shrinks into a single point at $K$ and then expands forming a Dirac point at about $E_B = 45 \pm 10$ meV.

In Fig. 3a, b, we show the ARPES data along $\Gamma$–$K$–$M$–$K$ with photon energies from 70 to 180 eV, which covers near four BZs along $k_z$. In Figure 3a, one can see that the band dispersions show no qualitative change at various photon energies. Especially, the $\beta$ band shows no noticeable $k_z$ dependence, which is also identified by the constant crossing point of $k_{F,\parallel}$ in Fig. 3b. The FB1 at about 0.4 eV below $E_F$ is also presented in every photon energy, with a limited bandwidth (does not exceed 150 meV) along $\Gamma$–$A$ as shown in Fig. 3d, indicating a near 2D character of the band. Figure 3b shows the integrated intensity at $E_F \pm 10$ meV in $\Gamma K$–$AH$ plane which covers part of $K$–$M$–$K$ at extended in-plane BZs. The $\beta$ band shows a little variation, further confirms the 2D-like character and ensures the 2D Dirac cone of DP1. The Dirac-related $\alpha$ band around $\bar{K}$ also displays a negligible dispersion along $k_z$, which has intensity modulation associated with matrix element effect, similar to FeSn and CoSn[25,29]. The $k_z$ dependence supports the quasi-2D characters of the DP1, DP2, and FB1 bands. An explanation to the 2D-like band structure is that the bands mainly originate from the orbitals confined by the kagome lattice.

To confirm the speculation of the orbital characters in the helical magnetic state along $c$-axis, we carry out the DFT + DMFT calculations in FM configuration with SOC and show the results in Fig. 3e (and in Supplementary Fig. 4). In the DFT + DMFT calculated spectra, we downshift the chemical potential of 76 meV to match the experiment value $E_{exp}$ mostly caused by a chemical doping of the sample. The calculations in the FM

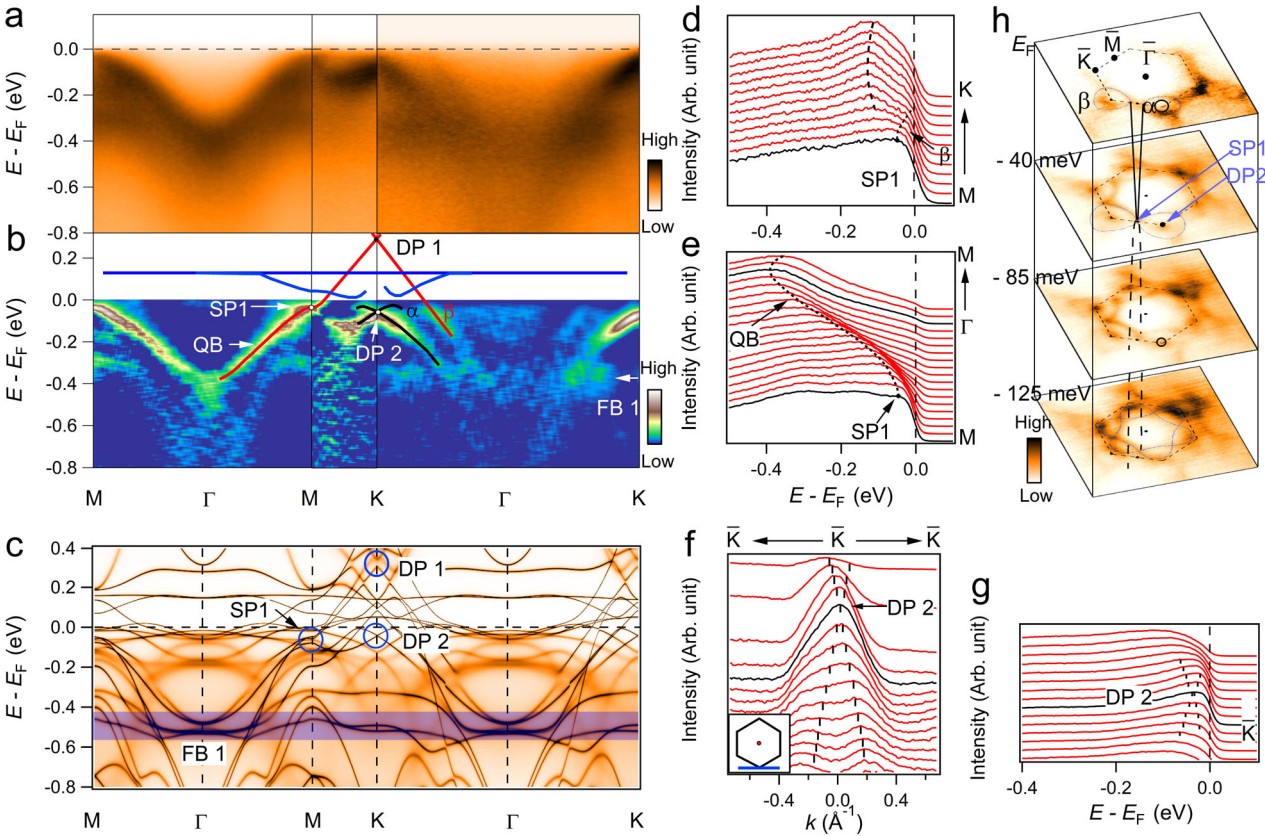

**Fig. 2 Band structure evolution in the 1st BZ. a** Photoemission intensity plots of YMn$_6$Sn$_6$ along Γ–M–K–Γ in the $k_z$ ~ 0 plane. **b** Corresponding second derivative plots of **a**. The appended colored lines serve as guides to the bands, which are extracted from DFT + DMFT calculations. **c** DFT + DMFT calculated ARPES in the FM state with SOC and with the experimentally determined $E_F$ shifted downwards about 76 meV. Dirac point (DP1 and DP2), saddle point (SP1) and flat band (FB1) are indicated by the blue circles and blue-colored region, respectively. **d, e** EDC plots along K–M and M–Γ–M, respectively. **f, g** MDC and EDC plots along the $\bar{K}$–$\bar{K}$–$\bar{K}$ direction, with the momentum path indicated as inset. **h** Constant energy maps at different binding energies.

configurations agreed well with the observation, the orbital-resolved ARPES without SOC shows that the β band mainly originates from the minority-spin branch of $d_{xy}/d_{x^2-y^2}$ orbitals (Supplementary Fig. 5). The DP1 locates at ~0.3 eV above $E_F$ and shows negligible dispersion along K–H as shown in Fig. 3e. The FB1 passing through the whole in-plane BZ originates from the minority-spin branch of the in-plane $d_{xy}/d_{x^2-y^2}$ orbitals, and has a limited bandwidth at the entire BZ district.

**Signature of phase-destructive flat band near $E_F$.** A feature that is faint in the 1st BZ but more clearly visible in the 2nd BZ is the existence of another flat band near $E_F$ at the BZ center, labeled as FB2 in Fig. 4. In Fig. 4a, the ARPES intensity plots and the corresponding second derivative plots along the Γ–M–K–Γ lines of the 2nd BZ at $k_z$ ~ 0 plane are shown. In the 2nd BZ, a spectral weight close to $E_F$ with binding energy ~60 ± 20 meV is clearly seen and extends over a large part of the BZ except around the K points due to the intensity leakage from the Dirac bands. In comparison, the intensity of the β and FB1 bands are stronger in the 1st BZ but become feeble in the extended BZ. We assign the contrasting behaviors between FB2 and FB1+β bands to the Brillouin-Zone-selection effects that show the different signal intensity in the first BZ and extended BZ[36,37] due to the different parity, symmetries or spin-polarization of the bands. In Fig. 4b, we present the DFT + DMFT calculations of the band structure in the FM state with SOC, with a downshift adjustment of the chemical potential of 76 meV as in Fig. 3e. Figure 4c, d show EDC

plots over more than one BZ along the high-symmetry line as marked in the insets. The contrast of the intensity among different BZs is shown, and the flat bands (FB1 and FB2) and the Dirac point (DP2) can be confirmed unambiguously. For the FB1, it is predicted to degenerate with the QB at the center of the BZ (Γ) without SOC. With the consideration of SOC, the two bands further hybridize and open a gap ~40 ± 10 meV, which is similar to the results in PM CoSn[28]. The FB2 declines weakly close to M and K points. It is noticed that the peak width becomes broader and the intensity becomes weaker near M and K points, which cause the expected Dirac points feature blurry and nearly indistinguishable, in agreement with the DFT + DMFT calculations in Fig. 4b. In conventional DFT calculations, there should be more band features in magnetic state, and several kagome-related structures are expected. However, in the DFT + DMFT calculations with consideration of the correlation effect which is normally present in the magnetic system, the broadness and the weaker intensity cause the smearing out of the band, the ARPES measurements normally observe fewer bands than DFT predictions, i.e., some band features around M, K, and along M–K.

We also present the evolution of the FB2 as a function of out-of-plane momentum $k_z$ measured in the in-plane-2nd-BZ along Γ'–A' as shown in Supplementary Fig. 3. The FB2 displays a weak dispersion over more than one BZ along $k_z$. Combining its in-plane flat dispersion in the $k_z$ ~ 0 plane, weak dispersion along $k_z$, and the DFT + DMFT calculated orbital-resolved ARPES as shown in Supplementary Fig. 5, we assign the orbital character of FB2 to $d_{xz}/d_{yz}$ orbitals. It is worth notice that there exists an

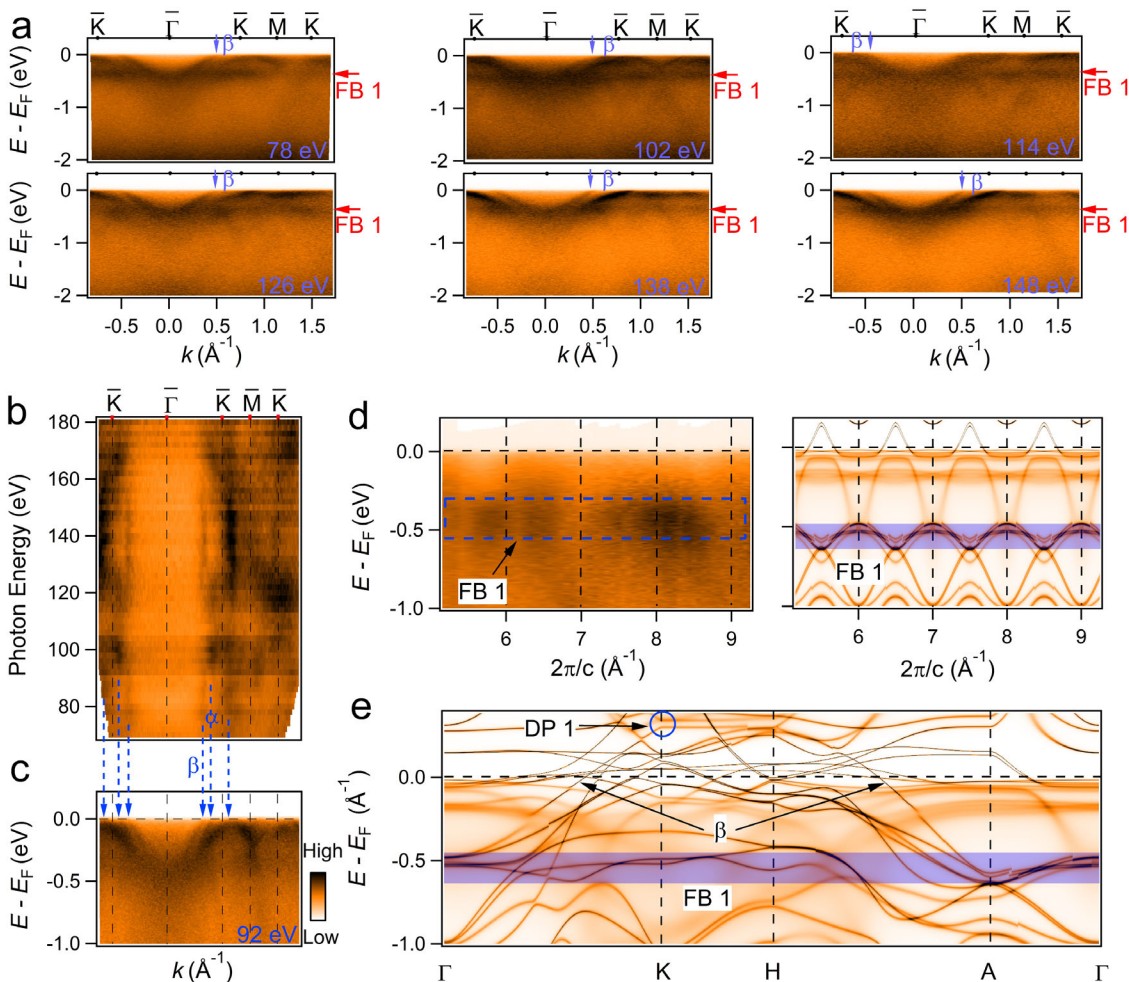

**Fig. 3 Photon-energy-dependence measurement of YMn₆Sn₆. a** Photoemission intensity plots of YMn₆Sn₆ with variable photon energies along $\Gamma(A)$–$K$ ($H$), the flat band (FB1) locates at the binding energy of about 0.4 eV through the whole BZ. The $\beta$ band appears in every photon energy. **b** ARPES intensity map at $E_F$ in the $k_\parallel$-$k_z$ plane with variable photon energies along $\Gamma(A)$-$K(H)$, showing a well defined 2D-like band structure. **c** Energy-momentum dispersion of YMn₆Sn₆ measured at $h\nu = 92$ eV ($k_z \sim 0.6~\pi/c$). **d** Photoemission intensity plot and DFT + DMFT calculated ARPES in the FM state with SOC along the $\Gamma$–$A$ direction, respectively. **e** DFT + DMFT calculated ARPES in the FM state with SOC along $\Gamma$–$K$–$H$–$A$–$\Gamma$, with the experimentally determined $E_F$ shifted downwards about 76 meV. The blue-colored region highlights the manifestation of the kagome flat band.

out-of-plane $d_{z^2}$ orbital with a steep $k_z$ dispersion and crossing $E_F$ at certain $k_z$, it thus might contribute to the $c$-axis conductivity in our transport measurements. Further, the FB1 displays a limited bandwidth (<150 meV) along $k_z$ in the 2nd BZ as shown in Supplementary Fig. 3. It is consistent with results in the 1st BZ with the in-plane $d_{xy}/d_{x^2-y^2}$ orbital composition, as discussed earlier. According to the spin-polarized DFT + DMFT calculations, FB2 with a high density of states (DOS) around the $E_F$ is from the majority-spin state, indicating its singly spin degenerate origin. FB2 is the first momentum-space evidence of the flat band really close to $E_F$ in the magnetic kagome system, which could give interesting phenomena such as orbital magnetism[32].

**The band calculations.** To take into account the strong electronic correlation effect of the Mn 3$d$ electrons, in the DFT + DMFT calculations, we include an onsite Coulomb interaction parametrized with a Hubbard $U = 4.0$ eV and a Hund's coupling $J = 0.7$ eV among the Mn 3$d$ electrons in both the PM and FM states. In the PM state, the mass enhancements of the Mn 3$d$ electrons near the $E_F$ are about 5–7, which are similar to the values in some iron chalcogenide superconductors[41]. The fluctuating local moment of Mn 3$d$ electrons, namely, the average value of $g[S(S$

$+1)]^{1/2}$, is about 3.9 $\mu_B$ with an effective spin $S = 1.5$, indicating YMn₆Sn₆ has a large fluctuating local moment due to the Hund's rule coupling. Combining with its metallic behavior, we conclude that YMn₆Sn₆ is a strongly correlated Hund's metal[41].

When the Mn 3$d$ local moments are partially frozen and form long-range static FM order in the hexagonal $ab$ plane, the mass enhancements of the Mn 3$d$ electrons near the $E_F$ are substantially reduced to about 2-3. The fluctuating local moment of Mn 3$d$ electrons remains the same value as in the PM state whereas the statically ordered moment is 2.1 $\mu_B$, which is in excellent agreement with experimental measurement[33,34].

## Discussion

In the PM state, the YMn₆Sn₆ has two major characters, including the strong electronic correlation and the kagome related features. The kagome structure has a flat band, while the strong correlation gives extra mass enhancement. These combinations will contribute large DOS around $E_F$ and could cause several instabilities, such as charge density waves[42], superconductivity[12,43], or magnetic instability[9,44]. In the magnetic state, the spin degeneracies are lifted, and a few spin-polarized branches shift below $E_F$. ARPES data reveal the existence of flat band with large DOS and Dirac point near $E_F$ in YMn₆Sn₆, these bands are thus of singly

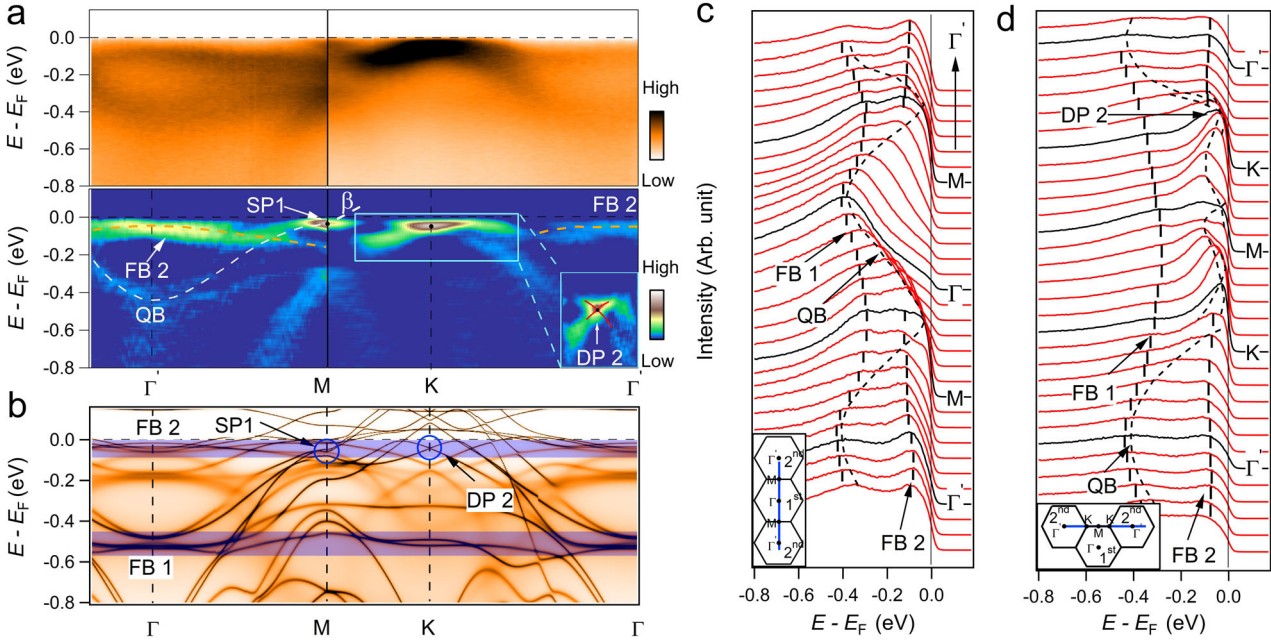

**Fig. 4 Band structure evolution in the 2nd BZ. a** Photoemission intensity plots and corresponding second derivative plots of $YMn_6Sn_6$ in 2nd BZ along $\Gamma$–$M$–$K$–$\Gamma$ in the $k_z \sim 0$ plane. The dashed lines are served as guides to the eyes. Inset: with the zoomed-in plot. **b** DFT + DMFT calculated ARPES in the FM state with SOC to according to the corresponding path of **a** and with experimental $E_F$. Dirac point, saddle point and flat band are indicated by the blue circles and blue-colored regions, respectively. **c**, **d** EDC plots along the high-symmetry lines, with the momentum paths indicated as inset. The flat bands (FB1 and FB2) and the parabolic band (QB) are indicated by the black dashed lines with thick and thin, respectively.

spin degenerate branches. With consideration of SOC, Chern gaps are opened and Chern numbers are assigned to each band correspondingly.

$YMn_6Sn_6$ posses a ferromagnetic bilayers which antiferromagnetically couple with neighboring bilayers. At low temperature, it exhibits an incommensate magnetic phase along $c$-axis[33,35]. With the breaking of the combined symmetry of inversion and time-reversal symmetry $PT$, the electron bands become spin-polarized. In order to confirm the spin configuration of individual bands and the corresponding flat band and Dirac point (Weyl point), further spin-polarized ARPES measurements are needed.

These spin-polarized bands carry Berry curvatures and also cause the orbital magnetism. The existence of an orbital magnetic moment has been reported in $Co_3Sn_2S_2$, and was attributed to the kagome flat band[32]. However, the ARPES observation of flat band near $E_F$ has not been reported in $Co_3Sn_2S_2$ yet. The orbital magnetism of the flat band of tight-binding model in kagome lattice with Kane–Mele SOC was calculated in Supplementary Fig. 7, which is closely related to the Berry curvature. Both the flat band with non-zero group velocity part and the massive Dirac fermion will contribute to the orbital magnetism. Our ARPES results first reveal FB and Dirac fermion near $E_F$ with orbital-selective characters, so multiorbital-magnetisms are expected.

In summary, based on ARPES measurement and in combination with theoretical calculations, we have fully revealed the band structure of magnetic kagome $YMn_6Sn_6$, and presented the first experimental observation of the complete characteristics of kagome lattice near $E_F$ with spin polarization and non-trivial topological properties. The Dirac point and flat band near $E_F$ arise from the spin-polarized band with intrinsic Berry curvature may explain the anomalous Hall effect observed in transport measurements, and the orbital magnetic moment observed in STM/S measurement. As an ideal candidate for magnetic kagome lattice material with the electronic structure near $E_F$, it opens up a new

avenue to comprehend the intrinsic properties of magnetic topological electronic material. Furthermore, if the non-trivial band structures–Dirac points, flat band and/or saddle point are further tuned properly, it would possibly realize more versatile quantum phenomena in such material.

## Methods

**Sample growth and characterizations.** Single crystals of $YMn_6Sn_6$ were grown by using Sn flux. Y lumps (purity 99.99%), Mn granules (purity 99.9%), and Sn grains (purity 99.99%) with a molar ratio of Y:Mn:Sn = 1:6:30 were put into an alumina crucible and sealed in a quartz ampoule under partial argon atmosphere. The sealed quartz ampoule was heated up to 1273 K and held for 24 h. Then it was cooled down slowly to 873 K at a rate of 5 K/h. Finally, the ampoule was taken out from the furnace and decanted with a centrifuge to separate $YMn_6Sn_6$ crystals from excess Sn flux. Magnetization and electrical transport measurements were carried out by using Quantum Design PPMS-14 T.

**ARPES measurements.** ARPES measurement were performed at the Dreamline and 03U beamline of the Shanghai Synchrotron Radiation Facility (SSRF), and 1-squared ARPES end-station of BESSY. The optimal energy and angular resolutions were set to 20 meV and 0.2°, respectively. Samples were cleaved in situ along (001) surface. During the measurements, the temperature was kept at 25 K and the pressure was maintained less than $5 \times 10^{-11}$ Torr.

**DFT + DMFT calculations.** The electronic structures of $YMn_6Sn_6$ were computed by using DFT + DMFT[45]. The DFT part is based on the full-potential linear augmented plane wave method implemented in WIEN2k[46]. The Perdew–Burke–Ernzerhof generalized gradient approximation[47] is used for the exchange correlation functional. DFT + DMFT was implemented on top of WIEN2k and was described in details[48]. In the DFT + DMFT calculations, the electronic charge was computed self-consistently on DFT + DMFT density matrix. The quantum impurity problem was solved by the continuous time quantum Monte Carlo method[49,50] with a Hubbard $U = 4.0$ eV and Hund's rule coupling $J = 0.7$ eV. The experimental crystal structure[51] (space group P6/mmm, No. 191) of $YMn_6Sn_6$ with lattice constants $a = b = 5.512$ Å and $c = 8.984$ Å was used in the calculations.

## Data availability

The authors declare that the main data supporting the findings of this study are available within the paper and its Supplementary Information files. Extra data are available from the corresponding authors upon request.

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

## Acknowledgements

This work was support by the National Natural Science Foundation of China (NO. 11774421, U1875192, 11674030, 12074041, 11774423, and 11822412), the National Key R&D Program of China (Grants NO. 2016YFA0302300, 2016YFA0401002, 2017YFA0403401, 2018YFE0202600, and 2016YFA0300504), the Fundamental Research Funds for the Central Universities, and the Research Funds of Renmin University of China (RUC) (18XNLG14, 19XNLG17). Z.P.Y., G. W., and Z.H.Y. were supported by the Fundamental Research Funds for the Central Universities (Grant No. 310421113). The ARPES experiments were performed on the Dreamline beamline of SSRF and supported by the CAS Pioneer Hundred Talents Program. Part of this research used Beamline 03U of the SSRF, which is supported by ME2 project (11227902) from NSFC. The calculations used high performance computing clusters at BNU in Zhuhai and the National Super-computer Center in Guangzhou.

## Author contributions

Z.L., Z.P.Y., H.L., and S.W. provided strategy and advice for the research. M.L., W.S., R.L., Z.L., Y.H., and S.W. performed the ARPES measurements. Z.P.Y., G.W., and Z.H.Y. performed the theoretical calculations. Q.W., and H.L. synthesized the single crystals. All authors contributed to the manuscript.

## Competing interests

The authors declare no competing interests.
