## [Peer Review File · Nature Communications]

REVIEWER COMMENTS

Reviewer #1 (Remarks to the Author):

This manuscript presents study of electronic structure of a Kagome magnet YMn_6Sn_6 using ARPES and DFT calculations. Authors report observation of Dirac cones, saddle points and a flat band. RMn_6Sn_6 is a very interesting and promising family of materials at the intersection of magnetically frustrated systems, ferromagnetism and topology. These topics are of great recent interest within condensed matter physics community. To the best of my knowledge this is first ARPES data directly showing features of the electronic structure in this family of materials. Observations reported here are important for verifying DFT calculations and understanding of the fascinating physics at the intersection of magnetism and topology. I'm happy to recommend publication of this manuscript in Nature Communications. I would encourage authors to address following points when revising the manuscript.

- 1) How does measured Fermi surface compare to calculated one? There are no plots of DFT Fermi surface - reader may benefit from such plots. I would encourage adding calculated FS for both bulk and slab/surface calculations.
- 2) The bands measured in ARPES are very broad even at E_f , given very low residual resistivity (10^{-5} Ohm cm). Can authors comment on the origin of this broadening - is it intrinsic and related to magnetism or due to surface/ARPES effects?
- 3) the DP2 point in Fig. 4 is not very clearly resolved, in second derivative it looks more like a flat band just below E_f . Is it possible to divide the data by Fermi function or plot zoomed area to show this feature in more convincing way?
- 4) The data in supplementary Information (Fig S2) looks much better and cleaner than data in main text - authors may consider swapping those.
- 5) "Perspective" plots in Fig. 3a are not very informative, I would suggest to plot these data as regular 2D panels.

Reviewer #2 (Remarks to the Author):

The manuscript by Li et al. reports angle-resolved photoemission spectroscopy (ARPES) and first-principles band-structure calculation study on a kagome magnet YMn_6Sn_6 which consists of alternately stacked honeycomb Sn layers and kagome Mn layers. Through the band-structure mapping in 3D momentum space using photon-energy variable synchrotron sources, they show evidence for the Dirac cone and flat bands predicted in the kagome lattice. They further relate spin-polarized Dirac-cone-band dispersion to the anomalous Hall effect in transport measurements. In my opinion, the data reported and discussed here are timely, the ARPES experiments were carefully performed, and the manuscript is well written. However, I found that some of the authors' key statements are not well supported by the experimental data in the current version. Also, some overstatements need to be corrected. My specific comments are the following.

- 1) Although the authors strongly suggest the spin-polarized Dirac cone, for example, in the title of the manuscript, they conclude this simply from the comparison of experimental data and calculations in the ferromagnetic phase. To claim the spin-polarized Dirac cone, the authors need to show experimental evidence for it, e.g. by spin-resolved ARPES and/or circular-dichroism ARPES. If the

authors cannot provide such data, it is better to tune down throughout the manuscript the overstatement on the observation of spin polarization.

2) It is hard to recognize a flat band in Fig. 2a and 2b at 0.4 eV. It would be necessary to show existence of the flat band by plotting the EDCs in this energy range. The authors can just expand the energy range of the EDCs in Figs. 2d-f and trace the peak position of this flat band. Also, the readers may be confused about the existence of the flat band well below the Dirac cone because corresponding feature is absent in the tight-binding calculation in Fig. 1b.

3) It is difficult to see a linearly dispersive Dirac-cone band from the raw EDCs in Fig. 2f. In particular, one cannot clearly see the upper Dirac cone. Moreover, guidelines to follow the Dirac-cone dispersion is inconsistent between Fig. 2f and 2g. Authors need to be careful to insist the existence of Dirac-cone band. More careful data presentation and analysis around DP2 are required.

4) I could not see clearly the saddle point at the M point in Fig. 2. The corresponding band seems to continuously approach EF without forming "flat region" around the M point. The EF-crossing of this band across the M-K cut needs to be presented in more convincing way to show that the observed dispersion is indeed saddle-point-like (energy scale of Fig. 2d is too small to see the beta band)

Reviewer #3 (Remarks to the Author):

The authors presented detailed experimental ARPES characterization on the electronic structure of YMn₆Sn₆ in comparison with sophisticated DFT+DMFT calculations. It is confirmed that due to the underlying Kagome lattice, there exist flat bands, saddle points, and Dirac points. Importantly, in YMn₆Sn₆, such features are located close enough to the Fermi energy, offering possibilities for further manipulation.

The results collected are convincing, which mark a significant progress in searching and characterizing topological electronic structure in materials with Kagome lattices, particularly in magnetic materials. The manuscript will inspire further exploration of such materials from both experimental and theoretical aspects.

However, I would not suggest accepting the manuscripts in the current form as publication on Nat. Commun., as the following points should be clarified/elaborated.

(1) It has been experimentally confirmed that YMn₆Sn₆ adopts double fan spin structure below 326K. As the ARPES measurements are done at 25K, the question is whether the AFM helical magnetic structure would affect the electronic structure? For instance, due to the modulation generated by the helical magnetic configurations, the bands will get folded and hence the Dirac points will be broken.

Therefore, is there a chance to (a) carry out experimental ARPES measurement between 326 and 359 K with a different magnetic structure and (b) perform DFT+DMFT calculations with AFM (if not helical) magnetic structures to verify it?

(2) As a non-specialist on ARPES experiments, a naive question is why there are so many bands not visible in Fig. 1b in comparison to Fig. 1c?

(3) As the DFT+DMFT calculations are performed for the FM state, should not the Dirac point 2 be actually a Weyl point?

Also, it is suspected that the degeneracy of the "Dirac point 2" should be regulated by symmetry which depends on the magnetization directions, as the authors considered SOC in the calculations. Could the authors elaborate on this based on symmetry? In this regard, "a small bandgap <10 meV opens at DP2" might be reconsidered.

(4) In contrast to Fig. 1b, why the energy of the FB2 lower than that of the saddle point at M and also that of DP2 at K? Is there a reason why DP2 has the same energy as the saddle point at M, rather than separated from each other as sketched in Fig. 1b?

(5) The matrix element effect is mentioned at different places. Could the authors elaborate for this specific system how it would make the bands visible/not-visible? This would make the manuscript more readable.

(6) It is claimed that "this non-trivial Dirac Fermion is in the occupied state and closer to EF than in TbMn6Sn6, and contribute to the intrinsic anomalous Hall effect in transport measurement". How this can be true give the following point: (1) 45 meV is still a large energy distance and it is suspected that at K' there should be another Dirac/Weyl point of the opposite chirality (2) the real materials are with helical magnetic ordering thus topological Hall effect is expected rather than anomalous Hall effect?

(7) If I understand correctly, the theoretical magnetic moment of Mn is 3.9 Bohr magneton while the experimental value is 2.1. Why there is such a big difference?

(8) The final comment on orbital magnetization is particularly confusing. Why the Kane-Mele SOC is used in the tight-binding model? It is well known that Kane-Mele SOC is an effective off-site SOC while here there exist on-site SOC between d-orbitals.

(9) It is claimed that YMn6Sn6 is a strongly correlated Hund's metal. In this case, it is expected that the many-body renormalization of the d-orbitals should be different. Can the authors elaborate on this? Also, is nematic effect expected?

(10) Fig. 1c can be improved to make the crystal structure more visible.

Summary:

Thank all the reviewers for the comments and suggestions on our work. From the three reviewers' comments, they gave high comments on the novelty and significance of our work, with many advice on how to enhance the presentation of the paper. We have answered all the questions in detail and taking their advice to improve the manuscript. We think the current form much better in clarity and suitable for Nature communications.

In short summary, the ARPES work on kagome magnetic metals is still early stage in exploring their band structures and the related interesting phenomena. The structural/magnetic frustration, electron correlations in magnets make ARPES results not as sharp as in non-correlated topological systems. In theory aspect, how to deal with the magnetic correlations in kagome magnetic metals is a great challenge. Previous works either avoid supplying the DFT results or the agreement between DFT and ARPES is poor. Our DFT+DMFT method provides a very successful prediction and has yielded a good agreement between theory and experiments.

For the electronic structure in kagome metal, interesting phenomena have been reported, such as massive Dirac, Chern Dirac, magnetic instability. Recently, by tuning the saddle point close to EF, charge density waves and superconductivity have been reported in kagome metals CsV3Sb5 (PRL 125, 247002 (2020)). We think our finding of flat band and saddle point near EF would have further influence in studying the kagome system.

---Detailed response to all questions-----

To first Reviewer:

This manuscript presents study of electronic structure of a Kagome magnet YMn6Sn6 using ARPES and DFT calculations. Authors report observation of Dirac cones, saddle points and a flat band. RMn6Sn6 is a very interesting and promising family of materials at the intersection of magnetically frustrated systems, ferromagnetism and topology. These topics are of great recent interest within condensed matter physics community. To the best of my knowledge this is first ARPES data directly showing features of the electronic structure in this family of materials. Observations reported here are important for verifying DFT calculations and understanding of the

fascinating physics at the intersection of magnetism and topology. I'm happy to recommend publication of this manuscript in Nature Communications. I would encourage authors to address following points when revising the manuscript.

Reply:

Thanks for your comments and suggestions. We have answered the concerns in detail in the response.

1. How does measured Fermi surface compare to calculated one? There are no plots of DFT Fermi surface - reader may benefit from such plots. I would encourage adding calculated FS for both bulk and slab/surface calculations.

Reply:

In the current stage, the DFT calculations on the magnetic system are not working well and the Fermi surfaces from DFT (even DMFT) would be misleading.

The DFT calculation of kagome magnetic metals is of great challenge and the researchers suffered from the lack of agreement between the DFT calculations and experimental observations.

Up to now, the calculated FS and experimental observations do not agree well [*PRB* **102**, 155103(2020); *Nat.Mater* **11**,1090-1095(2017)]. And in many works, the calculated FSs are not presented [*Nature* **555**, 638-642(2018), *PRL* **121**, 096401 (2018), *Nat.commun* **11**, 4004(2020), *Nat. Mater* **19**, 163-169(2020), *Nature* **583**, 533-536 (2020)].

We speculate these could be due to the correlation effects and the complications from the magnetic structure. We included the DFT calculations of nonmagnetic state in the supplementary (Fig S6), the result is inferior to the DMFT calculations presented in the main text. With consideration of magnetism, the slab calculations become impractical in such complicate system.

We attribute it to the advantage of the DMFT method with the relatively well

consideration of the correlation effect. In the current stage, the DMFT calculation maybe is the most suitable method in studying the kagome metals.

2. The bands measured in ARPES are very broad even at E_f , given very low residual resistivity (10^{-5} Ohm cm). Can authors comment on the origin of this broadening - is it intrinsic and related to magnetism or due to surface/ARPES effects?

Reply:

The reviewer asked a very fundamental question in the ARPES measurement. Here we try our best to answer the comments and questions.

The residual resistivity (10^{-5} Ohm cm) is a good value but not an excellent one.

In ARPES measurement of the magnetic metals, especially with kagome structure, the broadening of ARPES spectra is very common and remains an obstacle in understanding many physical phenomena well.

Up to now, the ARPES measurements on magnetic systems yield very broad spectra, such as in Mn_3Sn [*Nat.Mater* **11**,1090-1095(2017)], $TbMn_6Sn_6$ [*Nature* **583**, 533–536 (2020).], and our result here. It might be related to intrinsic factors such as magnetic correlation and/or frustrated kagome structure. For example, both in magnetic ($TbMn_6Sn_6$ [*Nature* **583**, 533–536 (2020)]) and paramagnetic metal (YCr_6Ge_6 [arxiv:1906.07140]), the band structures are both very broad.

At the same time, the sample specialists are still working on the crystal quality in eliminating defects and domains, which extrinsically broadens the spectra. The surface/ARPES results will also broaden the spectra, such as the combination of the exposed surface from $MnSn$ and YSn layers. In current state, due to the complication from the magnetism and kagome structure, we cannot make a conclusion about the origin of this broadening yet. It deserves further detailed studies with the improvement of experimental technique and sample quality.

3. the DP2 point in Fig. 4 is not very clearly resolved, in second derivative it looks

more like a flat band just below E_f . Is it possible to divide the data by Fermi function or plot zoomed area to show this feature in more convincing way?

Reply:

Thanks to the reviewer to point out the possible confusion in our data presentation. The first presentation is on a large scale, and the feature looks flat.

We have zoomed in the energy scale around the DP2, and added it to the inset (Fig. 4). The presentation shows better visibility about the existence of DP2, which is also limited by the broadening of the spectra as mention in question 2. In combination with the EDCs, MDCs, intensity plot and second derivative method (an image enhancement method), we think we prove the existence of the Dirac point here.

4. The data in supplementary Information (Fig S2) looks much better and cleaner than data in main text - authors may consider swapping those.

Reply:

Thanks for the suggestion, we move the figure from supplementary to main body and changed the description correspondingly.

5. "Perspective" plots in Fig. 3a are not very informative, I would suggest to plot these data as regular 2D panels.

Reply:

Thank the reviewer for the suggestion, we changed the figure presentation from perspective to regular 2D panels to be more informative.

To second Reviewer:

The manuscript by Li et al. reports angle-resolved photoemission spectroscopy (ARPES) and first-principles band-structure calculation study on a kagome magnet YMn_6Sn_6 which consists of alternately stacked honeycomb Sn layers and kagome Mn layers. Through the band-structure mapping in 3D momentum space using photon-energy variable synchrotron sources, they show evidence for the Dirac cone and flat bands predicted in the kagome lattice. They further relate spin-polarized Dirac-cone-band dispersion to the anomalous Hall effect in transport measurements. In my opinion, the data reported and discussed here are timely, the ARPES experiments were carefully performed, and the manuscript is well written. However, I found that some of the authors' key statements are not well supported by the experimental data in the current version. Also, some overstatements need to be corrected. My specific comments are the following.

Reply:

Thanks for the comments on our experiments. For the statements the reviewer concerns, we answered in detail in the following replies, modify the figures to improve the data presentation and revise the manuscript correspondingly to soft the statement on spin-polarization.

1. Although the authors strongly suggest the spin-polarized Dirac cone, for example, in the title of the manuscript, they conclude this simply from the comparison of experimental data and calculations in the ferromagnetic phase. To claim the spin-polarized Dirac cone, the authors need to show experimental evidence for it, e.g. by spin-resolved ARPES and/or circular-dichroism ARPES. If the authors cannot provide such data, it is better to tune down throughout the manuscript the overstatement on the observation of spin polarization.

Reply:

We agree with the reviewer that to confirm the spin-polarized Dirac cone in such a ferromagnetic state the more conclusive way is better than simply comparing the non-spin-polarized ARPES with calculations. Considering the limitation of experiments, we will follow the reviewer's advice to tune the statement and soft out claims. Thanks for the suggestion, and preciseness of the reviewer.

In the current stage, most of the ARPES measurements on magnetic systems yielded not very satisfying ARPES results in clearance, visibility of bands, and resolutions. Also, the agreement between calculations in magnetic states and observations is not satisfying, maybe due to some fundamental difficulties in the system. We have listed the difficulties of achieving better spectra in response to question 2 to the first reviewer.

Up to now, the data quality we obtained and presented in the manuscript is one of the highest qualities in 166-systems, comparing with ferromagnetic TbMn6Sn6 (Nature 583, 533–536 (2020), Fig 3 and Fig S7) and paramagnetic YCr6Ge6 (arxiv:1906.07140.)

It is essential to use spin-resolved ARPES measurement to fully confirm the spin nature of the band. However, considering the efficiency of the spin-determination and the broadness of the spectra, much more effort will be taken in the future.

We take the reviewer's comments and tune the statement about the spin polarization in the manuscript.

2. It is hard to recognize a flat band in Fig. 2a and 2b at 0.4 eV. It would be necessary to show existence of the flat band by plotting the EDCs in this energy range. The authors can just expand the energy range of the EDCs in Figs. 2d-f and trace the peak position of this flat band. Also, the readers may be confused about the existence of the flat band well below the Dirac cone because corresponding feature is absent in the tight-binding calculation in Fig. 1b.

Reply:

Q1: We take reviewer's advice. We have EDCs plots over several BZs and mark

the flat bands in Fig 4. Due to the limitation of 2D image, the EDCs plot is more convincing (Fig 4c, 4d).

Q2: In the tight-binding model of a Kagome system, the sign before the hopping term could be positive or negative. The flat band at 0.4 eV and the corresponding saddle point/Dirac point is consistent with a positive sign. We will change the parameter in figure 1b (from negative to positive) to reduce the confusion.

3. It is difficult to see a linearly dispersive Dirac-cone band from the raw EDCs in Fig. 2f. In particular, one cannot clearly see the upper Dirac cone. Moreover, guidelines to follow the Dirac-cone dispersion is inconsistent between Fig. 2f and 2g. Authors need to be careful to insist the existence of Dirac-cone band. More careful data presentation and analysis around DP2 are required.

Reply:

Taking the advice, we make more careful data presentation and analysis in the revision. We change the energy scale range and momentum range to make the DP2 better visible (Fig 2f, 2g). It also is further supported in extended BZ (inset of revised Fig 4a). We will revise the corresponding text to make the claim clear.

4. I could not see clearly the saddle point at the M point in Fig. 2. The corresponding band seems to continuously approach EF without forming “flat region” around the M point. The EF-crossing of this band across the M-K cut needs to be presented in more convincing way to show that the observed dispersion is indeed saddle-point-like (energy scale of Fig. 2d is too small to see the beta band)

Reply:

Taking reviewer’s advice, we have changed the energy scale in Fig 2d-2g to make the presentation more convincing. The saddle point and beta band are better visualized in EDC plots (Fig 2d, 2e).

To third Reviewer:

The authors presented detailed experimental ARPES characterization on the electronic structure of YMn_6Sn_6 in comparison with sophisticated DFT+DMFT calculations. It is confirmed that due to the underlying Kagome lattice, there exist flat bands, saddle points, and Dirac points. Importantly, in YMn_6Sn_6 , such features are located close enough to the Fermi energy, offering possibilities for further manipulation.

The results collected are convincing, which mark a significant progress in searching and characterizing topological electronic structure in materials with Kagome lattices, particularly in magnetic materials. The manuscript will inspire further exploration of such materials from both experimental and theoretical aspects.

However, I would not suggest accepting the manuscripts in the current form as publication on Nat. Commun., as the following points should be clarified/elaborated.

Reply:

Thanks for the highly comments on our results and the future influence. We have answered the concerns in detail in the following about the theoretical aspect. For some concerns in the ARPES bands, we made detailed explanations, and suggestions for future experiments, we will keep working in the near future although it is challenging.

1. It has been experimentally confirmed that YMn_6Sn_6 adopts double fan spin structure below 326K. As the ARPES measurements are done at 25K, the question is whether the AFM helical magnetic structure would affect the electronic structure? For instance, due to the modulation generated by the helical magnetic configurations, the bands will get folded and hence the Dirac points will be broken.

Therefore, is there a chance to (a) carry out experimental ARPES measurement between 326 and 359 K with a different magnetic structure and (b) perform

DFT+DMFT calculations with AFM (if not helical) magnetic structures to verify it?

Reply:

The reviewer's questions are very fundamental and he/she is pointing out the future efforts the field needs to work on.

Up to now, the understanding of the electronic structure in metallic magnetism is still at the early stage. The practical difficulties lie in several aspects: (1) the agreement between calculations and experimental observations still needs improvements. We are happy that in this system, we have reached some agreements between theory and ARPES measurements. (2) In theory, the sensitivity of the band structure to the spin configurations remains a great challenge. In experiments, the changes caused by the spin configuration are also overtaken by the resolution effect of the current experimental setup.

It deserves further work in answering the reviewer's question, and quite a lot of progress in the field is needed before we can answer the question. In the experimental part, the "high" temperature measurements (much less than 326-359K) in such a system are broad and blurry, and the data at room temperature is just smeared. In theoretical calculations, as we mentioned in the previous response to the first and the second reviewer, the agreement between DFT and experimental data is not satisfying, and we thus did DMFT to incorporate the correlations and achieved a relatively well agreement. Considering the complication of the magnetism and the correlation effects, the computational resource is not enough for DFT+DMFT with a large structure of few YMn_6Sn_6 unit cells yet.

2. As a non-specialist on ARPES experiments, a naive question is why there are so many bands not visible in Fig. 1b in comparison to Fig. 1c?

Reply:

The reviewer asked an important fact on the ARPES technique, a very fundamental but complicated question. In ARPES measurement, often the

observed bands are less than the calculations.

Several factors are credited to the missing of the calculated bands. (1) The matrix elements effect will tune the intensity of APRES bands as a function of electron states, light polarization, electron orbital symmetry, and binding energies. Under few not all possible conditions, it is normal to observe only part of the calculated bands. (2) The calculation produces band structure at $T=0\text{K}$ mostly, and the thermal broadening, scattering process will broaden the feature to some extent to indistinguishable. (3) The band structures from the calculation are oversimplified, which overestimate the intensity of many bands, such as the shadow bands in superlattice should be much weaker than a simply equal-weighted folding scenario.

Thus, we are happy to observe some main features of the kagome structure, the flat band, the saddle points at M, and some hints of the Dirac bands. And further, we are even happier to observe the flat band feature near EF in the extended Brillouin Zone which is most likely tuned by the matrix elements effect. These observations are already predicated by the DMFT calculations.

3. As the DFT+DMFT calculations are performed for the FM state, should not the Dirac point 2 be actually a Weyl point?

Also, it is suspected that the degeneracy of the "Dirac point 2" should be regulated by symmetry which depends on the magnetization directions, as the authors considered SOC in the calculations. Could the authors elaborate on this based on symmetry? In this regard, "a small bandgap $<10\text{ meV}$ opens at DP2" might be reconsidered.

Reply:

Indeed, with the breaking of the time-reversal symmetry, the commonly called Dirac point in kagome lattice in the literature should be Weyl point (Nature 583, 533–536 (2020), Nature 555, 638–642(2018)), we follow the convention to call it Dirac point. Without considering SOC, the corresponding two bands cross K point at the same energy and give rise to a Weyl point. After considering SOC, the band crossing is avoided and opens a $\sim 10\text{ meV}$ gap in the DFT+DMFT calculations.

4. In contrast to Fig. 1b, why the energy of the FB2 lower than that of the saddle point at M and also that of DP2 at K? Is there a reason why DP2 has the same energy as the saddle point at M, rather than separated from each other as sketched in Fig. 1b?

Reply:

Sorry for the confusion in the assignment of the number. The FB2 and the saddle point at M are from different orbital components. The FB1, SP at M, and DP1 (from theoretical calculation and extrapolation of the band from ARPES) at K are of one set. The DP2 is from another origin, with its flat band capping at above EF. Then the DP2, and SP at M with the same energy is just a coincidence.

In the revision, the FB1, SP at M, and DP1 are changed to FB1, SP1 at M, and DP1 to remove the confusion.

5. The matrix element effect is mentioned at different places. Could the authors elaborate for this specific system how it would make the bands visible/not-visible? This would make the manuscript more readable.

Reply:

This relates to questions as reviewers asked. The matrix elements effect is mentioned in several places and was attributed to the missing band in observations, and the enhancement of the band in the extended Brillouin Zone.

For a rough estimation of the matrix elements effect, there have been many papers. Such as PRB 85, 214518 (2012), and dark valley effect on graphene (PRB 77, 1195403, 2008), and Brillouin-Zone-selection effects (PRB 51, 13614 by F. Himpsel).

We add few sentences about the matrix elements (“due to the Brillouin-Zone-selection effects, the FB2 band shows almost no signal in the first BZ but enhances at the extended BZ”, PRB 51, 13614, 1995) from the reviewer’s suggestion to make it more readable to the general reader.

6. It is claimed that "this non-trivial Dirac Fermion is in the occupied state and closer to EF than in TbMn6Sn6, and contribute to the intrinsic anomalous Hall effect in transport measurement". How this can be true give the following point: (1) 45 meV is still a large energy distance and it is suspected that at K' there should be another Dirac/Weyl point of the opposite chirality (2) the real materials are with helical magnetic ordering thus topological Hall effect is expected rather than anomalous Hall effect?

Reply:

We thank the third Reviewer for this comment. First, at the high magnetic field, the spins in YMn6Sn6 will become fully polarized (ferromagnetic state), which would lead to the intrinsic anomalous Hall effect (AHE). This intrinsic AHE has been observed in the transport measurement [PRB 103, 014416 (2021)]. Second, although the position of the Dirac point is 45 meV away from the EF, it does not mean that the Dirac point can not lead the AHE. For example, for Fe3Sn2 and TbMn6Sn6, the Dirac points in these systems are 70 meV and 130 meV away from the EF, however, a large AHE still can be observed [Nature 555, 638 (2018); Nature 533, 583 (2020)]. The key point is that if the EF is exactly in the gap of Dirac point, we will get the quantized anomalous Hall conductivity per kagome layer [Nature 555, 638 (2018); Nature 533, 583 (2020)]. When the EF is moving away from the Dirac point, the anomalous Hall conductivity will not disappear suddenly and it will just decrease gradually. Thus when $E_B = 45$ meV, we still can observe the AHE in YMn6Sn6 at the high magnetic field. Third, we agree with the third Reviewer and the AHE can not be observed in the helical antiferromagnetic state at zero field, and the topological Hall effect is observed indeed at the low-field region once the spin-flop transition appears [PRB 103, 014416 (2021)]. In order to avoid the ambiguity, we have changed the sentence in the revised manuscript into "In our result, this non-trivial Dirac fermion is in the occupied state and closer to EF than in TbMn6Sn6, and contributes to the intrinsic anomalous Hall effect at high magnetic field in transport measurement".

7. If I understand correctly, the theoretical magnetic moment of Mn is 3.9 Bohr magneton while the experimental value is 2.1. Why there is such a big difference?

Reply:

The magnetic moment of Mn is 3.9 Bohr and 2.1 Bohr are both theoretical values. With $U=4.0$ eV and $J=0.7$ eV, the DFT+DMFT calculated fluctuating magnetic moment of the Mn atom is $3.9 \mu_B$ in both the paramagnetic state and the ferromagnetic state. In the ferromagnetic state, part of the fluctuating Mn magnetic moment becomes frozen to form a long-range static magnetic order. The DFT+DMFT calculated static magnetic moment in the ferromagnetic state is $2.1 \mu_B$, in excellent agreement with experiment (2.20 Bohr, PRB(R) 101, 100405 (2020)).

8. The final comment on orbital magnetization is particularly confusing. Why the Kane-Mele SOC is used in the tight-binding model? It is well known that Kane-Mele SOC is an effective off-site SOC while here there exist on-site SOC between d-orbitals.

Reply:

In the kagome system like YMn_6Sn_6 , five 3d orbitals are close and more than one orbital is extended and close to EF.

The Kane-Mele SOC is then used in the effective tight-binding model to describe the itinerate electrons.

9. It is claimed that YMn_6Sn_6 is a strongly correlated Hund's metal. In this case, it is expected that the many-body renormalization of the d-orbitals should be different. Can the authors elaborate on this? Also, is nematic effect expected?

Reply:

Indeed, the many-body renormalization of the Mn d-orbitals are different. With $U=4.0$ eV and $J=0.7$ eV, the DFT+DMFT calculated mass enhancement are about 4, 6, 7 for the Mn dz^2 , dx^2-y^2/dxy , dxz/dyz orbitals respectively in the paramagnetic state. In the ferromagnetic state, with the freezing of part of the fluctuating moment, the corresponding mass enhancement are reduced to about 2.8, 2.4, 2.5 in the majority spin channel, and about 2.3, 2.1, 2.1 in the minority spin channel.

In our measurement, from the consistency between DMFT and ARPES, we got a many-body renormalization of 2.1 for the band with QB/SP1/DP1. In order to determine the many-body renormalization values of other d-orbitals, the band information above EF is needed and beyond the scope of this manuscript.

The nematicity has been reported by STM results on Fe₃Sn₂ [Nature 562, 91–95(2018)], but it is hard to identify in ARPES currently. Here, the nematic effect is not considered in the DFT+DMFT calculations. We would like to check if any hints of the nematic effect in this material in future works.

10. Fig. 1c can be improved to make the crystal structure more visible.

Reply:

Thanks for pointing out and we modify the figure to improve the visibility.

REVIEWERS' COMMENTS

Reviewer #1 (Remarks to the Author):

Authors responded to all my queries in satisfactory manner and made revisions that improved the manuscript. In my opinion the reply and changes suggested by other two referees are also satisfactory. I'm very happy to recommend the publication of the revised version in Nature Communications.

Reviewer #2 (Remarks to the Author):

I appreciate the authors for their efforts to incorporate suggestions from the reviewers. I think that their answers to my specific concerns, in particular, regarding the overstatement of the spin polarized nature of Dirac cone, existence of flat bands, and effective presentation of linear Dirac-cone dispersion, are satisfactory. In my opinion, the manuscript has been appropriately revised. I would recommend the publication of this manuscript in Nature Communications in its present form.

Reviewer #3 (Remarks to the Author):

The authors have addressed the problems I asked with appropriate modifications implemented, thus I would recommend accepting the manuscript for publication on Nature Communications.

Nevertheless, though I would again that the Kane-Mele SOC model can be used to demonstrate the underlying physics as regulated by symmetries, I still think a more rigorous tight-binding model is needed for a quantitative description. Anyway, this is a minor aspect, not the main focus of the current draft.